# Improving the Contrast of Pseudothermal Ghost Images Based on the Measured Signal Distribution of Speckle Fields

Zhe Sun [1,2,*] , Frederik Tuitje [1,2] and Christian Spielmann [1,2]

1 Abbe Center of Photonics, Institute of Optics and Quantum Electronics, Friedrich Schiller University, Max Wien Platz 1, 07743 Jena, Germany; frederik.tuitje@uni-jena.de (F.T.); christian.spielmann@uni-jena.de (C.S.)
2 Helmholtz Institute Jena, Fröbelstieg 3, 07743 Jena, Germany
* Correspondence: zhe.sun@uni-jena.de

**Featured Application: The speckles' pixel intensity distribution in pseudothermal ghost imaging can influence the measured contrast-to-noise ratio (CNR) on a microscale. Based on these findings, it will be possible to develop an optimized light source that can pave the way for high-quality pseudothermal ghost imaging in a wide wavelength range.**

**Abstract:** In this study, we examine the quality of microscale ghost images as a function of the measured histographic signal distribution of the speckle fields from a nonuniform pseudothermal light source. This research shows that the distribution of the detected signal level on each pixel of the camera plays a significant role in improving the contrast-to-noise ratio (CNR) of pseudothermal ghost imaging. To our knowledge, the scaling of CNR with different pixel intensity distributions of the speckle fields is observed for the first time in the field of pseudothermal microscale ghost imaging. The experimental observations are in very good agreement with numerical analysis. Based on these findings, we can predict the settings for light sources that will maximize the CNR of microscale ghost images.

**Keywords:** ghost imaging; speckle field; pixel intensity distribution; contrast-to-noise ratio

## 1. Introduction

Optical microscopic imaging is a versatile and widespread tool in modern life science, fundamental physics, and chemistry. Exploring and breaking the optical imaging limitations, e.g., resolution and contrast, is the primary task. Exploiting the quantum properties of light is one way to overcome some of those limitations. Ghost imaging is one of the subfields of quantum imaging that exploits quantum or spatial intensity–fluctuation correlations to image objects with resolution, contrast, or other imaging criteria that can go beyond classical optics [1–3]. Ghost imaging can nonlocally image an object using photons that have not interacted with the object [4]. Theories and experiments have shown that both entangled photon interference and classical intensity–fluctuation correlations could be used for ghost imaging. In 1995, the first successful ghost imaging experiment relied on entangled photon pairs generated by spontaneous parametric downconversion [5]. This experiment was the first use of light containing spatial quantum correlations to illuminate an object to be imaged in quantum imaging. Soon afterward, Bennink et al. [6] presented an experimental demonstration of ghost imaging by using a pseudothermal light source. Strictly speaking, pseudothermal ghost imaging is no longer quantum imaging in the full sense, even if there are partial quantum correlations that exist in pseudothermal light. It is well known that the image quality of quantum ghost images is much better than in pseudothermal ghost imaging, but it is limited by wavelength [7–9]. Pseudothermal ghost imaging is very flexible regarding wavelength, e.g., X-ray ghost imaging [10–12]; thus, this technology has attracted more attention.

In recent years, ghost imaging with pseudothermal light sources has been investigated extensively [13–16] to overcome the limitations of image quality, which depends on detection contrast. Many strategies significantly improve the image quality in pseudothermal ghost imaging. One approach realizes a high-quality image by using high-order correlation [17–20]. However, it is hard to reach the theoretical limit of the contrast in high-order ghost imaging. The other important approach is implementing speckle-based imaging methods [21–24]. Recent results [25] have demonstrated the scaling laws for the achievable contrast of the retrieved ghost images, which strongly depend on the ratio between object size and the speckle size of the pseudothermal light for the same number of independent iterations. There are also various approaches focused on improving the imaging quality by using computational methods, e.g., sparsity constraints [26,27], THz patterns [28,29], differential ghost imaging [30], and computational ghost imaging by using special patterns [31–35]. Based on these studies, the improvement of pseudothermal ghost imaging quality has been greatly promoted. The listed studies generally concern the ghost imaging algorithm and the influence of special speckle patterns on ghost image quality. In microscale pseudothermal ghost imaging, the diffused light is weak and easily disturbed by stray light in detection. Thus, in addition to speckle patterns and size, the influence of the histographic pixel intensity distribution of the speckle fields must be considered; i.e., we have to evaluate the probability distribution of the normalized signal levels measured on a single pixel of our camera. To our knowledge, classical ghost imaging with different pixel intensity distributions of the speckle fields from a nonuniform pseudothermal light source has not been discussed yet.

In this paper, we report on the influence of the speckles' pixel intensity distribution from the nonuniform pseudothermal light source on the measured contrast-to-noise ratio (CNR) in microscale ghost imaging. We analyze the influence of the full width at half maximum (FWHM) and the peak positions of the Gaussian-like speckles' pixel intensity distribution to minimize the unfavorable influence caused by the nonuniform pseudothermal light field in experiments and simulations. This research has a great influence on microscale imaging. It can be used as a supplement to improve other methods or technologies. Our previous results on manipulating speckle pattern and speckle size, especially for high-quality microscopic ghost imaging, suggest that ghost imaging could be an interesting alternative for the imaging of radiation-damage-sensitive samples [36].

## 2. Theory

Pseudothermal ghost imaging relies on correlating a measured 2-D intensity speckle pattern to a transmission or reflection scalar value of an object illuminated by the same speckle pattern. By varying the speckle pattern over time and simultaneously measuring the transmitted light through the object, a statistical reconstruction of the image is possible. Thus, the measured 2-D speckle pattern is weighted with the corresponding scalar value of an object in every iteration and is added to the ghost image. The ghost imaging algorithm and the measurements for the quality of a ghost image used in this work were published in [19,20] and consider the nonlinear background and fluctuations of the light source given by

$$G\left(\vec{p}\right) = \left\langle \left( \frac{I\left(\vec{p}\right)}{\left\langle I\left(\vec{p}\right)\right\rangle} - 1 \right) \left( S - \frac{\langle S \rangle S_I}{\langle S_I \rangle} \right) \right\rangle, \tag{1}$$

where the ghost image $G\left(\vec{p}\right)$ with the spatial coordinate $\vec{p}$ depends on the measured intensity pattern $I\left(\vec{p}\right)$, the scalar value $S$, and the integrated intensity $S_I = \int I\left(\vec{p}\right)d\vec{p}$ of the object. The brackets represent the mean over all iterations.

The randomness of the rotating ground glass diffuser in pseudothermal ghost imaging will impose the field fluctuations that arise. The CNR is an important quantity that specifies the quality of an image and can represent the noise and fluctuations in the fields that illuminate the two detectors. For object signal and background noise in a ghost image

to be distinguishable, a good CNR value is necessary. Here, we use the CNR to measure the quality of a ghost image. The CNR is calculated from the object signal strength and the background signal strength and normalized to the image noise [37,38]:

$$CNR = \frac{\langle I_o \rangle - \langle I_b \rangle}{\sigma_{GI}}, \tag{2}$$

where $\langle I_o \rangle$ is the mean signal of the object area, $\langle I_b \rangle$ is the mean signal of the background in the ghost image, and $\sigma_{GI}$ is the standard derivation of the image considering noise, which is generally taken as that of the background.

### 3. Experimental Setup and Results

The experimental setup to study the evolution of the CNR with the pixel intensity distribution is shown in Figure 1. A helium–neon laser (632.8 nm) and a rotating diffuser in the focal plane of a lens constituted a nonuniform pseudothermal light source. Adding further elements into the beam path, we fully controlled the pixel distribution. With an adjustable neutral density filter, we increased or decreased the brightness of speckles' pixel intensity distributions to change the peak displacement. Inserting a second ground glass located before the diffuser allowed us to adjust the FWHM of speckles' pixel intensity distributions. Finally, by opening the aperture to change the beam waist and focal length of the lens, we ensured the same average speckle size in all experiments [21,22]. To exclude the influence of variation in speckle size on the CNR, we kept the scatter size to 30 μm in the experiment. The generated speckle field was divided by a beam splitter into two spatially correlated beams, namely the reference beam path and the object beam path. The object beam hit the object (letter "μ"), and this was followed by the integration of the transmitted light by a single-pixel detector, resulting in the so-called bucket signal. It is worth noting that the reference beam never interacted with the object, and its speckle pattern was recorded with a CCD camera having 1200 × 1600 pixels and an effective pixel size of 5.86 × 5.86 μm². Ghost imaging with randomly distributed fields works only for the same propagation lengths of the beam path from the beam splitter to the object and the beam path from the beam splitter to the CCD camera to conserve correlations. The propagation lengths' error between the beam paths was within a few millimeters. To avoid errors when measuring the CNR, we kept the position of the pseudothermal source and the measured target constant during the whole experiment and measured the same image area retrieved. The μ-shaped transmitting object was laser-cut in black-coated 100-μm-thick aluminum foil to ensure stability and to fully block light in the outer areas. Figure 1 also shows the designed object and the measured laser-cut result. The light transmitted the letter "μ". The width of the vertical bar of the letter "μ" in the object was 60 μm. Correspondingly, the transmitted area of the object was calculated to be ~77 × 10³ μm².

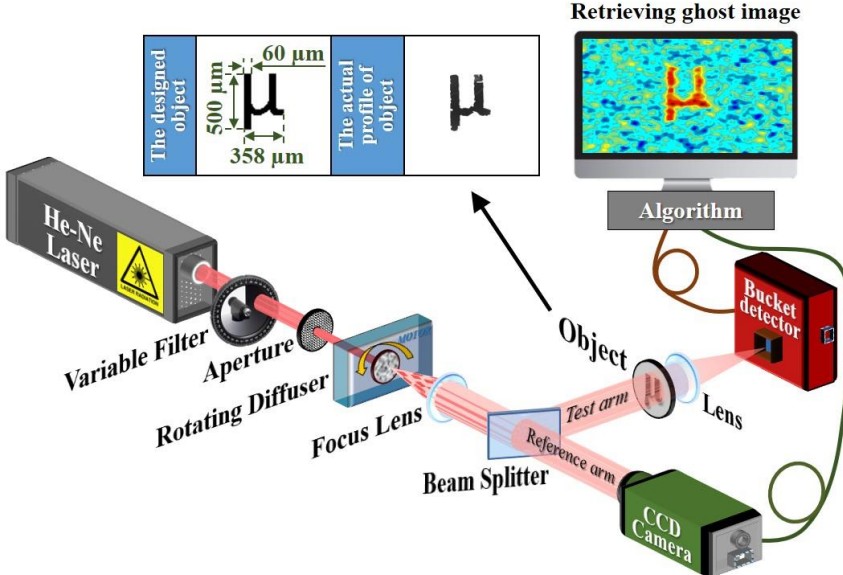

**Figure 1.** Schematic of the microscale pseudothermal ghost imaging experimental setup and the used object. Because of the laser-cutting fabrication process, the object size is certified by conventional microscopy.

The CNR of the ghost image relied on the speckles' pixel intensity distribution from the nonuniform pseudothermal light, which was detected by the CCD camera. The speckles' pixel intensity distribution was manipulated by different parameters in the experiment, as described above. To better understand and generalize the role of the speckles' pixel intensity distribution on the CNR of the retrieved ghost images for an object with a given size, we first evaluated the CNR as a function of the FWHM of the Gaussian-like speckles' pixel intensity distribution from the nonuniform pseudothermal light. Our camera had a dynamic range of 8 bits, so the maximum pixel count was 256. To generalize our findings, we normalized all pixel counts to the maximum count rate. In our experiment, the FWHM of the speckles' pixel intensity distribution was increased from 92 to 146 pixel intensity values in six steps with a fixed speckle size of 30 µm by reconstruction with 10,000 iterations under the same conditions. We can see from Figure 2d that the mean CNR of the object "µ" increased rapidly from 0.51 to 1.13 with an increase in the FWHM of the speckles' pixel intensity distribution. By contrast, after being modulated by corresponding optical elements, the FWHM of the speckles' pixel intensity distribution became wider, as we can see in Figure 2a. To compare the influence of the FWHM on the speckles' pixel intensity distribution, we normalized the pixel count to the maximum. The linear scaling of the retrieved CNR strongly depended on the FWHM of the speckles' pixel intensity distribution for the same number of independent iterations. The contrast and resolution can be seen in the horizontal section of the normalized Y-axis data of the vertical bar of the letter "µ" plotted in Figure 2c, where the experimental results of the CNR agree well with the evolution of the CNR in Figure 2d. As the FWHM of the speckles' pixel intensity distribution decreased, the CNR degraded gradually, as displayed in Figure 2b, as did the quality of the images. However, because of the well-known trade-off between resolution and CNR, the resolution did not show the same change in this case.

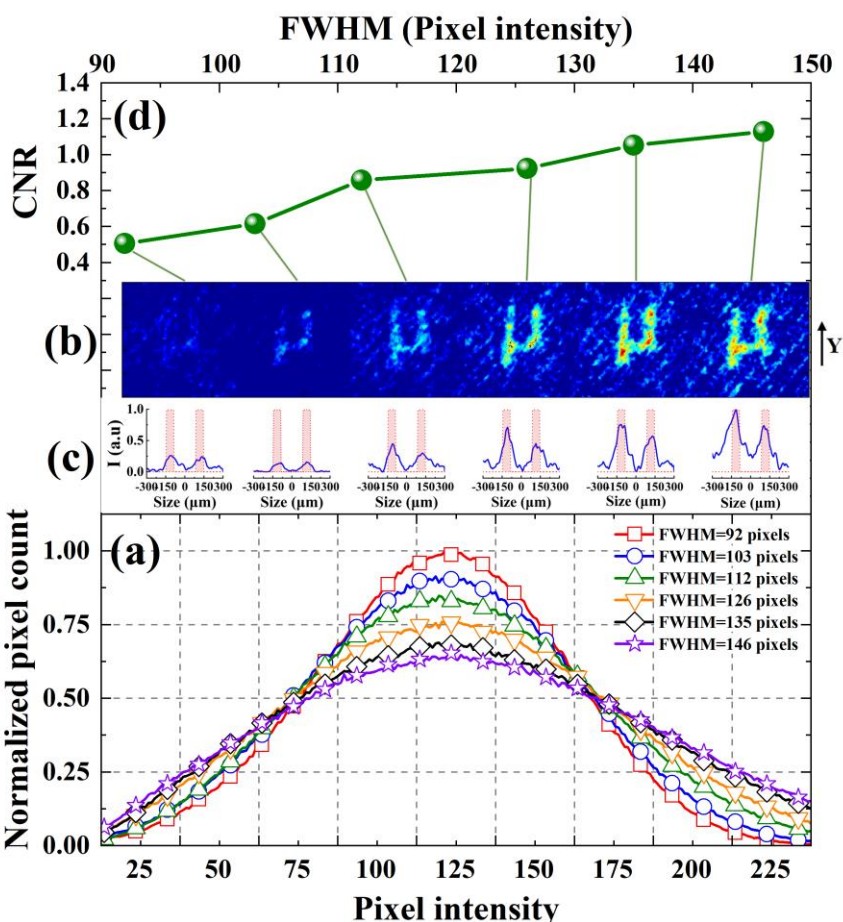

**Figure 2.** The evolution of the contrast-to-noise ratio (CNR) and reconstruction of the object image. (**a**) The measured speckles' pixel intensity distribution with different full widths at half maximum (FWHMs). Only one symbol is displayed for every 10 data. (**b**) The retrieved images of the letter "μ" in different FWHMs of the speckles' pixel intensity distributions. (**c**) Solid blue lines are the horizontal line-outs of the normalized data of the retrieved ghost images and red dashed curves display the object. (**d**) The CNR as a function of the FWHM of the speckles' pixel intensity distribution from the nonuniform pseudothermal light.

To have a better understanding of the CNR enhancement of ghost imaging by the speckles' pixel intensity distribution from the nonuniform pseudothermal light in our experiment, we recorded and retrieved images for different peak positions of the Gaussian-like speckles' pixel intensity distribution; the results are summarized in Figure 3. As can be seen in Figure 3a, when moving the position of the pixel intensity distribution peak from the pixel intensity of 80 first to 127, and then to 170, whilst ensuring nearly the same FWHM of the speckles' pixel intensity distribution, the image quality showed a sudden change. Figure 3b shows the image quality inspected by the naked eye. The quality increased from the left position of the speckles' pixel intensity distribution until it reached a maximum, at which point the speckles' pixel intensity distribution was at the center position, and then the quality dropped. As shown in Figure 3c, it was very easy to distinguish the horizontal section of the normalized Y-axis data of the vertical bar of the letter "μ" if the speckles' pixel intensity distribution was in the center position compared to others. Finally, we estimated the CNR as shown in Figure 3d, providing a quantitative confirmation of the qualitative inspection results. The corresponding CNR of the ghost image started from 0.43, reached a peak at 1.1, and then dropped to 0.7. Thus, the CNR was mainly determined by the FWHM of the speckles' pixel intensity distribution. Therefore, we came to the following conclusion: the best CNR of a ghost image can be achieved if the mean speckle pixel intensity is in the

center of the dynamic range of the CCD camera and the distribution has the maximum FWHM. The slight discrepancy may be because of some uncertainties in the FWHM of the speckles' pixel intensity distribution. It is hard to keep the FWHM the same in different peak positions of the speckles' pixel intensity distribution without adjusting the parameters of the CCD camera. At least our data prove that the widest FWHM at the center position of the speckles' pixel intensity distribution can achieve the best CNR for the given speckle size and object in microscale ghost imaging.

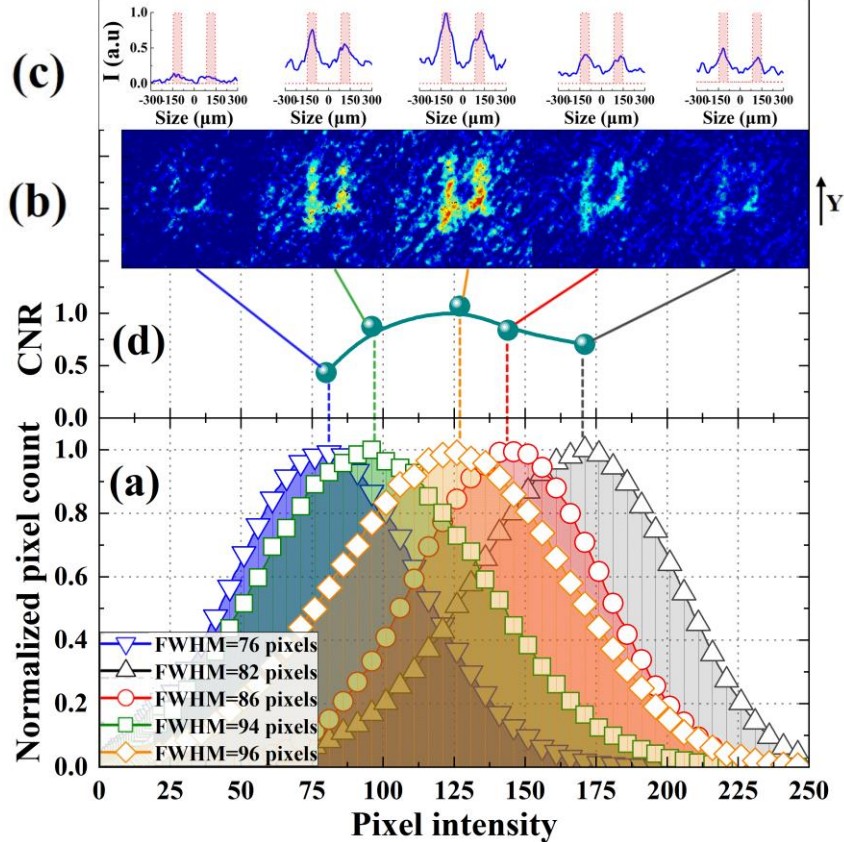

**Figure 3.** The evolution of the CNR and reconstruction of the object image. (**a**) The measured speckles' pixel intensity distribution with different peak positions. Only one symbol is displayed for every 5 data. (**b**) The retrieved images of the letter "μ" in different FWHMs of the speckles' pixel intensity distributions. (**c**) Solid blue lines are the horizontal line-outs of the normalized data of the retrieved ghost images and red dashed curves display the object. (**d**) The CNR as a function of different peak positions of the speckles' pixel intensity distribution from the nonuniform pseudothermal light.

The observed scaling depicted in Figures 2d and 3d, corresponding to Figures 2a and 3a, has never been reported in the literature. These experimental results reveal some interesting properties of the speckles' pixel intensity distribution on ghost imaging CNRs. This is the first hint that there are optimal FWHMs and center positions for the speckles' pixel intensity distribution to achieve the best CNR for a given speckle size and object in microscale ghost imaging. Hence, the speckles' pixel intensity distribution, which serves as the pseudothermal light field in the conventional ghost imaging system, gives a better contrast to the correlation imaging system. This scaling can also be easily verified when viewing the ghost images in false color representation in Figures 2b and 3b. In our experiment, the retrieved ghost images with the highest quality as judged with the naked eye were in good agreement with the images with the highest CNR.

## 4. Simulation Results

To validate the experimentally observed evolution of the CNR of a ghost image with the FWHM and the peak positions of speckles' pixel intensity distribution from the nonuniform pseudothermal light, a numerical model was developed to simulate the observed behavior. The numerical framework simulated the speckle pattern with a defined average speckle size and Gaussian-like distributions with a freely selectable width and position of maximum. The so-gathered speckle field $\Phi_{in}$ represented the speckle field at the object's position, shown in Figure 1. The speckle field was then multiplied elementwise by the object $O$ and numerically propagated to the photodiode's position, where the field was integrated to generate the bucket value $b = \int \Im(\Phi_{in} * O, d)$, where $\Im(\Phi, d)$ is a Fourier based complex field propagator for a field $\Phi$ over a distance $d$. The reconstruction itself was then done with the reconstruction code from [25]. The simulation framework then used a set of parameters for the width $\omega_h$ and the peak position of the maximum $p_h$ of the speckles' pixel intensity distribution to create a simulated ghost image $G(\omega_h, p_h)$ dependent on the distribution parameter set with a fixed average speckle size of 30 μm. As a parameter set, a width from 60 to 160 pixel intensity value and a position from the pixel intensity from 60 to 160 were chosen to include the experimental parameters. After generating the ghost image, the CNR was measured and plotted as a function of the parameter set to produce the CNR matrix shown in Figure 4. It was shown that the CNR rose with the increased width of the speckle field distributions, which corresponded to experimental observations. However, the dependence of the CNR on $p_h$ fell without showing a decent local maximum, which is explained by the shape of the distribution. The simulated field had a slightly asymmetrical Gaussian-like distribution of the histogram. When the pixel intensity distribution is simulated, there is a higher probability of such asymmetry. Shifting the maximum to brighter values led to saturation of more and more pixels because of the limited dynamic range and, therefore, to a loss of pixel intensity variety. A loss of pixel variety decreased the usable spatial information for image reconstructions and lowered the CNR. The local maximum of the experimental CNR-position dependence is explainable similarly. At low positions, zero-valued pixels became saturated. With an increasing position, this saturation decreased, and the pixel brightness variety rose. If the position increased further, the pixel saturation rose to high values again, leading to a lower CNR.

The simulation validates the experimentally found correlation between the quality of a ghost reconstruction and the distribution parameters of the input speckle field. Concerning the slightly deviating role of position dependence, an explanation involving the shape of the distributions, as they are not perfectly Gaussian, could be found.

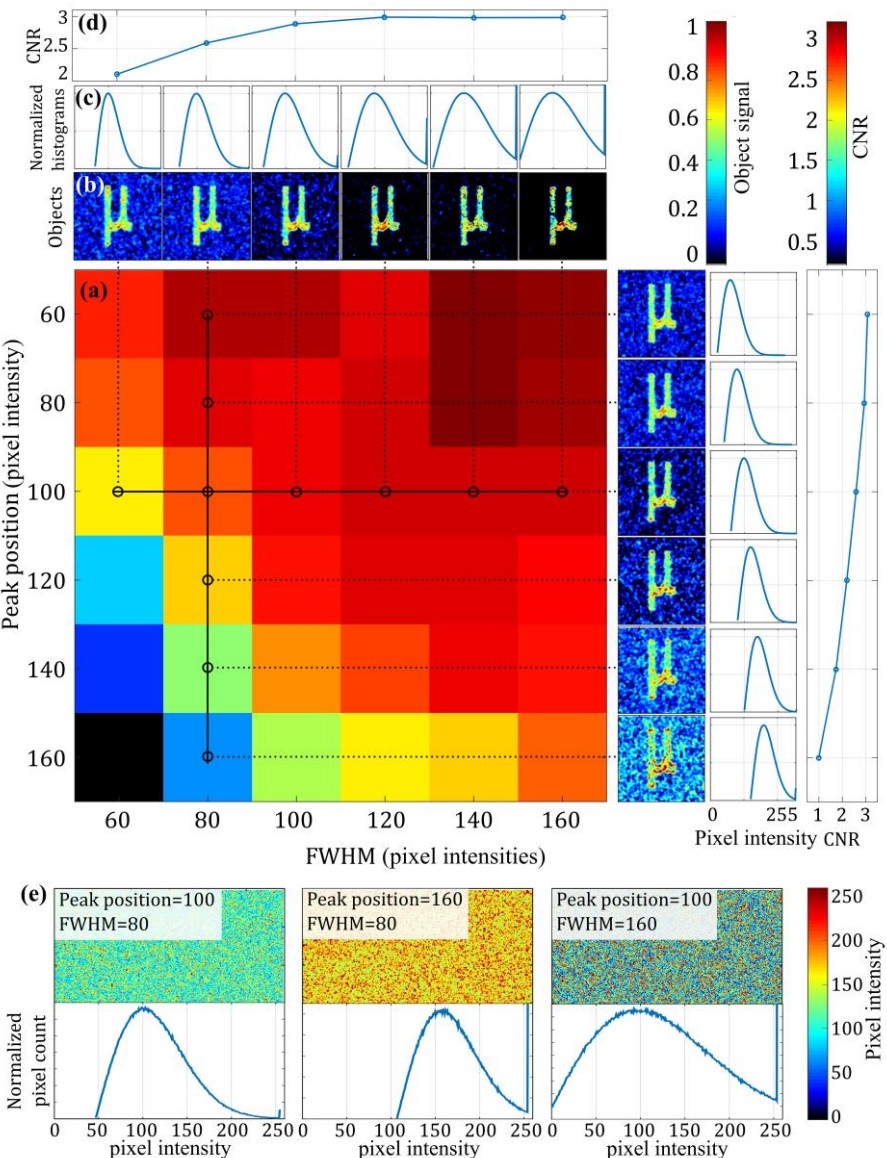

**Figure 4.** Results of the pixel intensity distribution dependent simulation. The CNR rises nearly continuously with the width of the pixel intensity distribution, whereas the CNR decreases with the increasing peak position of maximum (**a**,**d**). Parameters matching the experimental data are highlighted with black lines. The corresponding objects (**b**) and pixel intensity distributions (**c**) are plotted and connected with dashed lines. Exemplary speckle patterns for selected parameters (**e**) visualize saturation effects, which lead to information losses at an increasing position of maximum. On the other hand, a broader pixel intensity distribution provides more information for the algorithm to work with.

## 5. Conclusions

In conclusion, we have experimentally demonstrated the influence of the pixel intensity distribution of the speckle fields from the nonuniform pseudothermal light source on the measured CNR in microscale ghost imaging. The CNR of a ghost image is related to the speckles' pixel intensity distribution, which plays an important role in the CNR enhancement of ghost imaging. Thus, our experiments suggest that the CNR depends on the FWHM and the center position of the speckles' pixel intensity distribution.

Our simulation analysis unambiguously identifies the speckles' pixel intensity distribution as one of the major limiting factors in the growth of the CNR. Furthermore, it offers

a general approach applicable to all fields of imaging where a higher CNR is needed and can be applied.

Although we have not yet demonstrated the ghost imaging experiment in a relatively high CNR case with nanoscale resolution, it would be feasible by manipulating the speckle pattern and speckle size and optimizing the algorithm. Our study of the optimum speckle distribution in a wide range of parameters will be very important if ghost imaging will be implemented in wavelength regimes, e.g., in the extreme ultraviolet range (XUV), where the strong absorption of all materials reduces the possibility of manipulating light fields. Based on these findings it will be possible to develop an optimized light source that can pave the way for ghost imaging in the XUV. Ghost imaging in the XUV is very interesting because it will allow the taking of microscopic images with very high resolution whilst minimizing radiation damage.

**Author Contributions:** Data curation and writing—original draft preparation, Z.S.; methodology and formal analysis, F.T.; writing—review and editing, Z.S., F.T. and C.S.; supervision and funding acquisition, C.S.; All authors have read and agreed to the published version of the manuscript.

**Funding:** This research was funded by Deutsche Forschungsgemeinschaft (DFG, German Research Foundation) under Germany's Excellence Strategy—EXC 2051 (Project-ID 390713860), "Balance of the Microverse"; GSI Helmholtzzentrum für Schwerionenforschung GmbH & Postdoctoral Research Foundation of China, grant number GSI05HGF-GSI-OCPC-2017; and Federal State of Thuringia and the European Social Fund (ESF) Project, grant number 2018 FGR 0080.

**Data Availability Statement:** Data available on request due to restriction of privacy. The data presented in this study are available on request from the corresponding author. The data are not publicly available due to related experiments are still in progress and involve unpublished papers.

**Acknowledgments:** We thank Tobias Helk and Sukyoon Oh for their help with preliminary experiments and Brian Seyfarth for processing the samples.

**Conflicts of Interest:** The authors declare no conflict of interest.

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
