# Peer review of "Improving the Contrast of Pseudothermal Ghost Images Based on the Measured Signal Distribution of Speckle Fields"

_applsci, doi:10.3390/app11062621_

Round 1
Reviewer 1 Report
Dear Editor
this paper presents a detailed analysis of the dependence of contrast-to-noise-ratio in ghost imaging (GI) in function of speckles size distribution.
The analysis is rather technical, but it can find interest in the community working in the field.
The results look sound and are well presented.
Nevertheless, there are a few points to be improved.
- In the introduction the authors state that “This research has great influence on microscale imaging, even nanoscale…”. For an object to be imaged of 0.5 mm x 0.36 mm this claim looks excessive.
- The choice of CNR as figure of merit should be better motivated
- The CNR points in Fig.2-3 do not present uncertainties, why?
- The bibliography is limited (literature on ghost imaging is huge) and these results should be better inserted in the contest of present studies in quantum imaging. On the one hand, this can be reached by quoting a few review papers (e.g. Journal of Optics 18 (2016) 073002; Nature Reviews Physics 1, 367 (2019); …) and some significant papers on quantum imaging. On the other hand, it would be worth comparing the presented results with some connected analyses (e.g. Scientific Reports 2, 651 (2012); Phys Rev. A 83, 063807 (2011); Physics Letters A 376, 1519-1522 (2012);…)
- Finally, the main motivation adduced for this study is the application to GI with XUV light, while the experiment is done with red light. I think various motivations more could be adduced. The authors should improve the conclusions in this sense.
Once these questions are properly answered, this work could probably be accepted for publication.
Author Response
Response to Reviewer 1 Comments:
We thank the reviewers for their very careful reading and their comments and suggestion. We tried our best to answer the questions of reviewers and we hope the revised manuscript can be published now. The questions are answered one by one in following:
- In the introduction the authors state that “This research has great influence on microscale imaging, even nanoscale…”. For an object to be imaged of 0.5 mm x 0.36 mm this claim looks excessive.
Answer (A): For the revised version of the manuscript, we delete the “even nanoscale” in the last paragraph of the introduction part (marked in blue) to avoid misunderstandings. Our experiment was using a microscale sample (letter “μ” 500 μm×360 μm size with 60 μm width of the open area) for imaging. The nanoscale is just in reach, if we use shorter wavelengths in the XUV as discussed in the outlook. Line 68-71.
- The choice of CNR as figure of merit should be better motivated.
A: The reviewer’s suggestions are very constructive. About the choice of CNR, we add the descriptions and explanations in the following and theory part in the manuscript (marked in blue). There are two major categories of light sources used for ghost imaging: entangled photons produced in parametric down-conversion and pseudo-thermal light produced by a laser beam with a rotating diffuser or a spatial light modulator. Different from the entangled-photon ghost imaging, the pseudo-thermal ghost image always lies on a noisy background. Because the randomness of the rotating ground glass diffuser in the pseudo-thermal ghost imaging will impose the field fluctuations to arise. The CNR can represent the noise and fluctuations in the fields that illuminate the two detectors. Therefore, to better distinguish the two regions of interest in an image, we use CNR to measure the quality of a ghost image. Further CNR is widely used in the literature as a measure for the image quality. Line 87-97.
- The CNR points in Fig.2-3 do not present uncertainties, why?
A: We measured the CNR of the ghost images in Fig. 2 and 3 by using the same measure region in MATLAB. There are uncertainties by using different masks. But the scaling of the CNR is always consistent.
- The bibliography is limited (literature on ghost imaging is huge) and these results should be better inserted in the contest of present studies in quantum imaging. On the one hand, this can be reached by quoting a few review papers (e.g. Journal of Optics 18 (2016) 073002; Nature Reviews Physics 1, 367 (2019); …) and some significant papers on quantum imaging. On the other hand, it would be worth comparing the presented results with some connected analyses (e.g. Scientific Reports 2, 651 (2012); Phys Rev. A 83, 063807 (2011); Physics Letters A 376, 1519-1522 (2012);…)
A: We agree with the reviewer, that ghost imaging is sub-field of quantum imaging. Quantum imaging has been reviewed in many recent publications, which we have cited now. As mentioned by the referee, in this growing field it is almost impossible to give credit to all the previous work in the field of quantum imaging, especially in a short letter reporting on a specific experiment in ghost imaging. Nevertheless, we have added in the introduction part three significant papers on quantum imaging, including two papers suggested by the reviewer (marked in blue). Line 26-40, 53-56.
Ghost imaging is an emerging sub-field of quantum imaging. There are two types of ghost imaging. Type 1 ghost imaging utilizes quantum-entangled photons produced via SPDC and exploits quantum spatial correlations between the photon pairs that comprise the signal and idler output. Type 2 ghost imaging utilizes spatial intensity-fluctuation correlations between two spatially separated light fields to image objects. Strictly, the type 2 ghost imaging is no longer quantum imaging in the full sense, even if there are partial quantum correlations exist in pseudo-thermal light. In our manuscript, we focused on pseudo-thermal ghost imaging, which is classical imaging. It is unsuitable to present previous studies in quantum imaging. It is well known that the image quality of quantum ghost images is much better than classical ghost imaging, but limited by wavelength. Classical ghost imaging is very flexible in wavelength, e.g. X-ray ghost imaging, thus this technology has attracted more attention.
In our manuscript, we report on the influence of the speckles’ pixel intensity distribution from the nonuniform pseudo-thermal light source on the measured CNR in microscale ghost imaging. We propose and analyze a method or technology to improve the contrast of ghost imaging. It can be used as a supplement to improve other methods or technologies. Due to the different evaluations and methods of the quality of ghost imaging, it is hard to compare every method in improving ghost imaging quality. In our manuscript, we listed some typical methods in the second paragraph of the introduction part and revised some content (marked in blue) to explain our research aim by comparing the presented results.
We thank the reviewer to bring the mentioned papers to our attention. We have carefully read these publications and included them in our list of references if appropriate. Here are the detailed comments.
Physics Letters A 376, 1519-1522 (2012) investigated ghost imaging by sparsity constraints. This is an important approach focused on improving imaging quality. We mentioned in the second paragraph of the introduction part (ref. [16,17]). In our manuscript, we didn’t use this algorithm. In future experiments, we will consider incorporating this algorithm into our experiments.
Scientific Reports 2, 651 (2012) investigated the (microscopic) speckle-speckle correlations in Gaussian light sources used for ghost imaging. They showed that the quality of the imaging (SNR) was determined by the total correlations in the thermal source beams. Different from the microscopic speckle-speckle correlation properties, we focus on the macroscopic characteristics of the speckles’ intensity distribution.
Phys Rev. A 83, 063807 (2011) investigated a theory of SNR in bipartite ghost imaging with classical and quantum light. It analyzed the influence of the brightness of the source, the losses, the number of Spatio-temporal modes collected by the detector, and the resolution on performing ghost imaging in terms of SNR. They didn’t analyze the influence of the speckle field on SNR.
- Finally, the main motivation adduced for this study is the application to GI with XUV light, while the experiment is done with red light. I think various motivations more could be adduced. The authors should improve the conclusions in this sense.
A: Thank you for your suggestion. Our findings can be widely used in pseudo-thermal ghost imaging. I revised the Featured Application (marked in blue). XUV microscopic ghost imaging is the purpose and prospect of our project, which is one of the most important motivations. The XUV from a table-top setup is too weak to manipulate due to the strong absorption of all materials. Thus, it is necessary to study the optimum speckle distribution in a wide range of parameters. Line 10-13.

Reviewer 2 Report
This paper shows the influence of the width and peak of the light intensity distribution on the contrast-to-noise ratio of ghost images. This work is appreciable, so it deserves to be published but it seems written in a hurry and a lot of revisions are required according to the following hints.
General considerations
The term “pixel value distribution”, used throughout the paper, is not the most appropriate, according to me. Instead, I suggest to use “pixel intensity distribution” or “pixel luminosity distribution” because it is related to the amount of the light impinging on the CCD camera. Also the “pixel value” caption in figs. 2,3,4 should be renamed.
The plots in figs. 2a and 3a should be drawn with smaller or more separated symbols, because the curves are hard to recognize if the images are printed in black and white.
Specific revisions - the number(s) represent the corresponding line(s)
11 – The CNR acronym should be defined before using it.
107 – The calculated area probably refers to the surface that allows light to pass but it should be better defined.
124-125 – The words “from 0.51 to 1.13” refer to the increase of the CNR but it seems they refer to the FWHM of the distribution. They should be moved after the word “rapidly”.
125-126 – What does the sentence “By contrast….Figure 2(a)” mean? It seems that the conclusion is missing because the fact that the FWHM becomes narrower cannot be a conclusive remark.
147-148 – The sentence “By moving….” should be written “By moving the position of the intensity distribution peak from 80 to 170…”.
149-150 – The consideration “…when the pseudo-thermal…sudden change” is not clear. What do the authors mean when they write “the pseudo-thermal light has nearly the same FWHM as the speckles’ pixel value distribution”? Moreover, from fig. 3b and 3d no “sudden” change appears but a smooth variation of the CNR.
158-159 – “The corresponding CNR….drops to 0.27”. This is not true because the CNR increases from 0.43 to 1.1 and then goes down about to 0.7.
159 – The conclusion that the CNR is mainly determined by the FWHM of the intensity distribution seems to be correlated to fig. 2 rather than fig. 3. So, if it is a concluding comment, it should be moved when a global consideration is done, after line 177.
182-184 – “Hence,….gives an upper limit of the contrast”. This is not true, by observing the experimental data. In fact, in fig. 2d there is no upper limit and, in principle, the CNR could still increase if the FWHM was greater than 146.
188-190 – “To avoid errors…retrieved”. This notice should be given prior to any experimental discussion.
Section 4 – The authors speak about a “Gaussian-like distribution” but the shape of the simulated distribution recalls more a Poisson than a Gaussian one. It is not clear why the authors have not used a real Gaussian shape (such as the experimental shape they have observed) but this has a strong impact on the comparison with the experimental data, in particular for the lower distribution peak values. So, the authors should justify their choice.
214 – “…and a position from the pixel value of 60 to 160…” should be written “…and a position of the peak value from 60 to 160…”
216 – By the the word “independence” do you mean “as a function of”? Moreover, instead of “to receive” you should use “to produce”
220-221 –What do the authors want to say with the sentence “The simulated fields…values”? Maybe, they mean that their asymmetric distribution shows higher Y-values for higher X-values? In any case, this sentence should be re-written.
223-229 – This consideration is correct but it should already be mentioned when the experimental data are commented (in Section 3). Actually, the decrease in CNR when the intensity peak is very low or very high depends on the loss of the intensity variety, as the authors point out. The simulation is just a confirmation of this, apart from lower peak values where the simulated distribution is different from the experimental one and, contrary to the model, an experimental decrease does not appear.
236-238 – The sentence “Since the CNR…imaging” has not a conclusion. Please, check it.
Round 2
Reviewer 1 Report
Dear Editor
I think the authors have properly answered to all referees' questions, improving the paper. I suggest publishing the paper in its present form.